# A Robustly and Effectively Optimized Pre-training Approach for Masked Autoencoder

## Abstract

Recently, Masked Image Modeling (**MIM**) has increasingly reshaped the status quo of self-supervised visual pre-training. This paper does not describe a novel MIM framework, but to unravel several fundamental ingredients to robustly and effectively pre-train a Masked AutoEncoder (**MAE**) with improved downstream performance as a byproduct. We highlight the great significance for the whole autoencoder to encourage high-variance interactions across different tokens, while simultaneously for the reconstructed target to smooth the inter-patch variances. First, at the decoding phase, we apply the standard dropout upon the attention probabilities as noise to randomly mask out the edge connection across different tokens. Otherwise, their shortcut interactions might hinder the emergence of meaningful contextual representation. Second, we point out that the per-patch normalization will fail unless the patch pixels rely on some population statistics to reduce inter-patch variance and then smooth the reconstruction. Third, we show that autoencoders with different capacities encounter the issue to varying degrees and the learnable masked tokens can be employed to manipulate the variance dependent on its inserted position and ratio in the model. The proposed techniques here are simple and effective to benefit the pre-training of a masked autoencoder stably and obtain superior performance across different downstream tasks.

## 1 Introduction

Contrastive learning (He et al., 2020; Chen et al., 2020; Grill et al., 2020) and masked image modeling (Bao et al., 2021; He et al., 2022; Xie et al., 2022b) have become two dominant paradigms for self-supervised visual pre-training. This paper elaborates on the latter one, which exhibits more intriguing progress recently. Generally speaking, the key philosophy of masked image modeling is to mask out a portion of input image and then learn latent representation that can predict the removed data. Such mechanism has first manifested its efficacy in natural language processing (Kenton & Toutanova, 2019; Brown et al., 2020) to learn contextual representation that universally benefits various downstream tasks. Unfortunately, the vision community struggles to embark a similar trajectory for a while.

Thanks to the development of Vision Transformer (Dosovitskiy et al., 2020) (**ViT**), masked image modeling eventually opens the new chapter for self-supervised visual pre-training. In particular, the pioneering BEiT (Bao et al., 2021) applies ViT as a bidirectional encoder to predict visual tokens from a pre-trained codebook (Ramesh et al., 2021), given some patches of the input image are masked and replaced with a learnable embedding. BEiT first demonstrates the superiority of masked image modeling by outperforming the supervised version that pre-train using the class label of an image. Instead, He et al. (2022) presents a self-contained solution to directly reconstruct pixels at the decoding phase. Their proposed masked autoencoder employs an asymmetric architecture, where the encoder only computes on the low-portion visible tokens while the lightweight decoder is used to reconstruct the other high-portion masked tokens. Some similar techniques are also exploited in Xie et al. (2022b), such as random masking, raw pixel prediction, lightweight decoding, etc. In addition, Wei et al. (2022) reveals that using HoG (Dalal & Triggs, 2005) as the reconstructed target yields competitive representation.

In light of these breakthroughs, there arises several lines of improvements: incorporating with siamese-network-based contrastive learning (Huang et al., 2022; Tao et al., 2022); enhancing the

pre-trained image tokenizer (Zhou et al., 2021; Peng et al., 2022); etc. Instead, this paper does not fall within these categories, but to dive into a purely self-contained Masked AuoEncoder (MAE). Without loss of generality, we consider the architecture proposed in He et al. (2022) as our baseline. We attempt to reveal several fundamental ingredients that contribute to the success of masked autoencoding and then propose techniques to robustly and effectively improve it. The key message we deliver here is that it is of significance for the entire model to circumvent the oversmoothing token interactions at both encoding and decoding phase. Conversely, we show that some population statistics are in demand to smooth the inter-patch variance in the pixel space. Otherwise, the per-patch normalized pixels can not serve as a well-performing reconstructed target. We detail our key message as follows.

Particularly, when we talk about oversmoothing here, we mean that the pre-trained autoencoder might learn shortcut interactions across tokens to trivially fulfill the pretext task (e.g., pixel prediction), which hinders the emergence of contextual representation in masked image modeling. On the encoding side, both high-portion masking and random masking can alleviate this issue, which have been proposed in He et al. (2022). First, if we regard the self-attention matrix as a normalized adjacent matrix across the patches, then a complete connected graph will be created (Shi et al., 2022). To this end, the high-portion masking enables to dynamically sample a combinatorial number of subgraphs, substantially reducing the the risk of oversmoothing. Second, given a fixed masking ratio, the random masking will have larger bipartite entanglement between the visible and masked tokens by increasing their boundary perimeter. That is, to preserve a fixed global semantics, the random masking might be able to have higher masking ratio to spatially remove more redundant patches. While there have been effective practices for the encoding side, however, they are not applicable to the decoder. At the decoding phase, the removed tokens are inserted back as a shared learnable embedding and then all the tokens are visible to the decoder in a fully-connected graph. To circumvent the trivial dependencies among tokens, we are inspired by the drop edge technique (Rong et al., 2019) in graph neural network to randomly remove edges of the graph. In our specific case, we apply the standard dropout upon the self-attention probabilities to randomly mask out some portion of interactions across different tokens, dynamically resulting in a partial connected graph to be visible by the decoder at every iteration.

In addition to the architecture, what to predict (i.e., the reconstructed target) is also a concern. In He et al. (2022); Feichtenhofer et al. (2022), the per-patch normalization of the raw pixel is empirically demonstrated the optimal variant, which suggests that predicting the local high-frequency components benefit the representation learning. However, we argue that this operation becomes meaningless if some population statistics are missed to transform the pixel space. We conduct an ablation study to illustrate this point in Table 1, where the ViT-Base and ViT-Large models are pre-trained using different reconstructed targets. As the *None* variant shown in the table, if we directly reconstruct on the purely raw pixels with per-patch normalization, then the masked pre-training will not bring positive gains compared with training the model from scratch. Indeed, only if we first perform some specific inter-patch normalization, then followed by the intra-patch normalization, predicting the high-frequency components will be plausible. Specifically, the original MAE actually transforms the pixel space by normalizing along the RGB channels, where the mean and standard deviation are calculated image-wisely on the whole dataset. We also testify that the similar advantages can be observed by normalizing over the 1-D patches along the dimension of size equal to the patch length, which is shown in the last two columns in the table. Note that our conclusion is indeed aligned with Chen & He (2021); Wang et al. (2022), once we regard the predictor as an autoencoder and the original image as the naturally stop-gradient target view. To this end, performing batch normalization on the target will facilitate smoothing the target branch and thus stablize the pre-training.

While the previous discussions are delivered in an architecture-invariant manner, however, models with different capacities are subject to the mentioned issue by varying degrees. For instance, the extremely high masking ratio (e.g., 75%) might fit the ViT-Large model better but not the optimal one for the ViT-Base variant. In order to manipulate token interactions for different models, we show that the involvement of masked tokens could be more flexible in terms of their inserted position and ratio. For a smaller architecture, we can design an extra low-portion of mid-level masked tokens and include them earlier in the encoder. The inserted position is preferred at the higher layer of the encoder, as not to sacrifice the efficiency. Unlike the argument in Chen et al. (2022) that the representation learning and pretext task completion should not be coupled, we empirically demonstrate

Table 1: Top-1 finetuning accuracy (%) on ImageNet-1K with per-patch normalized pixels as the reconstructed target. If we directly perform intra-patch normalization on the raw pixels without first centering them based on some population statistics, then it will suffer from the significant performance drop.

| Architecture | Epoch | None | Global RGB Normalization | Global Patch Normalization |
|---|---|---|---|---|
| ViT-Base | 800 | 81.99 | 83.25 | 83.28 |
| ViT-Large | 800 | 84.70 | 85.29 | 85.47 |

that this manner does not hurt the performance of the encoder although it is optimized for the two tasks simultaneously.

## 2 METHOD

In this part, we first introduce the edge dropping of self-attention matrix at the decoding phase. Then, we will highlight the importance of conducting inter-patch normalization for the reconstructed target. Finally, we will show that the insertion of masked token can be more flexible in terms of ratio and position to manipulate the token interaction.

### 2.1 DROP EDGE AT THE DECODING

Our method relies on the vision transformer (Dosovitskiy et al., 2020) architecture, where the self-attention module serves as a principal element. Its computation can be simply shown in Equation 1

$$\text{Attention}(Q, K, V) = \text{softmax}\left(\frac{QK^T}{\sqrt{d_k}}\right) V, \tag{1}$$

where the $Q, K, V$ are a sequence of query vector, key vector and value vector that are obtained by multiplying every input token with some learnable matrices. Then the attention score is calculated by taking the dot product of every two query vector and key vector respectively to measure how closely there are interacting. In particular, this score will be divided by the square root of the dimension of the key vector (i.e., $d_k$) and then normalized by the softmax function to obtain a probability distribution. Eventually, the self-attention module will output the updated embeddings of the tokens by multiplying the value vector with the normalized probabilities and aggregating the results.

Our modification on the module is simply applying the dropout noise upon the attention probabilities to randomly mask out the interactions across different tokens, which can be represented as $\text{dropout}(\text{softmax}\left(\frac{QK^T}{\sqrt{d_k}}\right))V$. In fact, this operation has been existing in most of the public implementations of Transformer, Bert and Vision Transformer [1], etc, while is usually frozen in practice. Gao et al. (2021) utilizes this technique as a minimal data augmentation to generate positive pairs of sentence embeddings. In our specific case, we use it to imitate the edge dropping in graph neural network (Rong et al., 2019), dynamically presenting a partial connected graph to the decoder at very block and every step. To this end, the token embeddings are aggregated information dynamically according to a partial set of other tokens, which consequently capture the dependencies across tokens at various levels.

### 2.2 ON THE IMPORTANCE OF GLOBAL NORMALIZATION

As proposed in He et al. (2022), the intra-patch normalized pixels are proven to be the empirically optimal reconstructed target. Given a dataset with $N$ data points, suppose each input sample is spatially resized to $H \times W$ with three color channels, then the whole size of the dataset is $(N, H, W, 3)$. If we patchify the image into a sequence of $L$ tokens $\{x_{1...L}\}$, then each of them has $D$ dimensions,

---

[1] https://github.com/rwightman/pytorch-image-models/blob/master/timm/models/vision_transformer.py

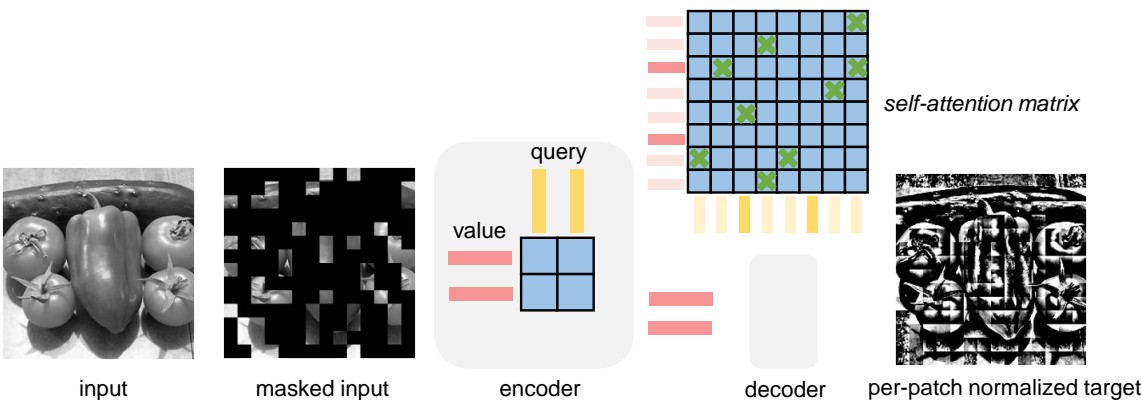

Figure 1: The general pipeline of masked image modeling. A high portion (e.g., 75%) of patches is first randomly masked out and only those visible patches are forwarded into a ViT-based autoencoder. The autoencoder is designed asymmetrically, where the decoder is highly lightweight than the encoder, in terms of depth and width, as the two gray boxes intuitively shown in the figure. At the encoding phase, only a low-portion patches are forwarded, which strongly sidesteps the spatial redundancy and encourages an effective context to be captured. However, since the masked tokens (marked with light color) are incorporated back to together interact with the visible tokens (marked with **dark color**), it increases the risk of oversmoothing. At this phase, we apply dropout on the self-attention matrix to randomly remove some interactions across different tokens, dynamically revealing a subgraph to the decoder. In the end, the per-patch normalized pixels appear empirically to be the optimal option for reconstruction, which enhances the contrast of high-frequency components. We further demonstrate that, to conduct this intra-patch normalization, we are supposed to first rely on some population statistics to reduce the variance across patches. Note that the sample image in the figure is illustrated in gray scale in order to refrain from the display problem of out-of-range values in the per-patch normalized version.

where $D = \frac{H}{\sqrt{L}} * \frac{W}{\sqrt{L}} * 3$. In this case, the dataset has the size of $(N, L, D)$. The per-patch normalization in He et al. (2022) calculates the mean $\mu_i \in \mathbb{R}$ and standard deviation $\sigma_i \in \mathbb{R}$ along the $D$ dimension for each patch $x_i$ as follows:

$$
\begin{aligned}
\mu_i &= \frac{1}{D} \sum_{j=1}^{D} x_i^j \\
\sigma_i^2 &= \frac{1}{D} \sum_{j=1}^{D} \left( x_i^j - \mu_i \right)^2,
\end{aligned}
\tag{2}
$$

where $x_i^j$ is the $j$-th pixel in the $i$-th token. To this end, each pixel within the patch is normalized as $\widehat{x}_i^j = \frac{x_i^j - \mu_i}{\sqrt{\sigma_i^2 + \epsilon}}$, where $\epsilon$ is a constant for numerical stability. As visualized in Figure 1, this manner enhances the contrast of high-frequency components within the patch, and thus casts the reconstruction a harder task, where a higher loss is empirically detected.

However, whether struggling to fit the high-frequency details is more intrinsic to obtain universal representation remains unclear. We find that this intra-patch normalization cannot work without certain condition, that is, pre-conditioning on some specific inter-patch normalization to first smooth patch-wise variances. Here we show an optional inter-patch normalization in Equation 3, where we pre-compute the patch-wise mean $\mu \in \mathbb{R}^D$ and standard deviation $\sigma \in \mathbb{R}^D$ on the whole dataset. Note that $x_{i,j} \in \mathbb{R}^D$ denotes the $j$-th patch in the $i$-th data point. Then, before conducting the

intra-patch normalization, every token $x$ will be first normalized as $\widehat{x} = \frac{x-\mu}{\sqrt{\sigma^2+\epsilon}}$.

$$
\begin{aligned}
\mu &= \frac{1}{NL} \sum_{i=1}^{N} \sum_{j=1}^{L} x_{i,j} \\
\sigma^2 &= \frac{1}{NL} \sum_{i=1}^{N} \sum_{j=1}^{L} (x_{i,j} - \mu)^2 .
\end{aligned}
\tag{3}
$$

### 2.3 THE FLEXIBLE INSERTION OF MASKED TOKEN

In the original masked autoencoder proposed in He et al. (2022), the encoder is fed with only the visible tokens, and a shared learnable embedding for all the masked tokens will not be incorporated until the decoding. However, it does not mean that we shall strictly abide by this practice. In fact, the insertion of masked tokens can be more flexible in terms of the ratio and position, which provides another option to manipulate the token interactions. Although a concern might be raised that the encoder are not supposed to simultaneously handle both the tasks of representation learning and pretext task completion (Chen et al., 2022), however, we empirically find that it does not hurt the encoding under this two-task setting. Since a fixed masking ratio might not be optimal for all the architectures with varying capacities, the masked token embedding can be exploited to encourage more token interactions for smaller model. For instance, the extremely high (e.g., 75%) masking ratio fits the ViT-Large model better but might be considerably higher for the ViT-Base variant. To address this issue, we similarly design a shared learnable embedding for representing another set of mid-level masked tokens (e.g., another 25%), compared with the bottleneck masked tokens in current masked autoencoder. In particular, we are only feeding these tokens into the relatively deep layers of the encoder, as not to introduce unaffordable degradation on the efficiency.

## 3 EXPERIMENTS

In this part, we first introduce our pre-training setup. Second, we will compare our pre-trained models with other methods upon the performance of different downstream tasks. Third, we conduct case studies to analyze the optimization loss with different reconstructed target and the positive gain brought from the flexible insertion of the masked tokens.

### 3.1 IMAGE CLASSIFICATION

**Pre-training Setup** Our experiments are conducted based on the official PyTorch implementation[2] of MAE (He et al., 2022). Specifically, we only rely on the database of ImageNet-1K (IN1K) (Deng et al., 2009) for the pre-training. Both ViT-Base (ViT-B/16) and ViT-Large (ViT-L/16) have been used to testify the efficacy of our proposed methods. We follow MAE to adopt its simple data augmentation. The input image is resized to 224×224 and split into a sequence of 196 patches, each of them has the patch size of 16×16. At the encoding phase, 75% of the patches are masked out and only the remaining 49 visible patches are fed into the encoder. At the bottleneck between the encoder and decoder, a shared learnable embedding for all the masked tokens are inserted back to interact with the visible tokens. We follow He et al. (2022) to employ a lightweight decoder and predict the pixels of the masked tokens. We randomly remove the edge interaction across tokens at the decoder with dropout probability of 0.3. Following Wei et al. (2022), we use [0.5,1.0] as the scale in random resized crop for ViT-Base model. If not specified, the other hyperparameters in the pre-training and fine-tuning recipes are simply inherited from He et al. (2022).

**Results** In Table 2, we compare our method with some typical works in the recent development of masked image modeling. From the table, we can draw the following conclusions: 1) Our 800-epoch pre-trained model can achieve the comparable performance against MAE's 1600-epoch counterpart, and thus significantly reduces the pre-training overhead; 2) Our model obtains considerable improvements over MAE in terms of the equal pre-trained epochs, which is marked with the gray rowcolor in the table; 3) Our model is directly modified from the MAE baseline, which shares the virtue of self-contained predicted target, simple augmentation view, training efficiency, etc.

---

[2]https://github.com/facebookresearch/mae

Table 2: Top-1 accuracy (%) on ImageNet using ViT-Base and ViT-Large.

| Method | Pre-train data | #Epochs | ImgSize (#views) | Top-1 Acc (%) |
|---|---|---|---|---|
| *Methods using ViT-B/16:* | | | | |
| scratch (He et al., 2022) | N/A | N/A | N/A | 82.3 |
| MoCo v3 (Chen et al., 2021) | IN1K | 300 | 224 (2) | 83.2 |
| DINO (Caron et al., 2021) | IN1K | 300 | 224/96 (2/10) | 82.8 |
| BEiT (Bao et al., 2021) | IN1K+DALLE | 800 | 224 (1) | 83.2 |
| MAE (our impl.) | IN1K | 800 | 224 (1) | 83.3 |
| MAE (He et al., 2022) | IN1K | 1600 | 224 (1) | 83.6/.5 (paper/git) |
| Ours | IN1K | 800 | 224 (1) | 83.6 |
| *Methods using ViT-L/16:* | | | | |
| scratch (He et al., 2022) | N/A | N/A | N/A | 82.6 |
| MoCo v3 (Chen et al., 2021) | IN1K | 300 | 224 (2) | 84.1 |
| BEiT (Bao et al., 2021) | IN1K+DALLE | 800 | 224 (1) | 85.2 |
| iBOT (Zhou et al., 2021) | IN1K | 1000 | 224 (2) | 84.8 |
| MAE (our impl.) | IN1K | 800 | 224 (1) | 85.3/.4 (/paper) |
| MAE (He et al., 2022) | IN1K | 1600 | 224 (1) | 85.9 |
| MaskFeat (Wei et al., 2022) | IN1K | 1600 | 224 (1) | 85.7 |
| Ours | IN1K | 800 | 224 (1) | 85.6 |

## 3.2 VISUALIZATION OF THE OPTIMIZATION LOSS

In Figure 2, we visualize the training loss curves with respect to different reconstructed targets, where each variant is pre-trained for 100 epochs. From this visualization, we can draw the following observations: 1) The optimizations across all the variants are converging stably; 2) In case the normalization is imposed on the raw pixels, it casts the reconstruction a harder task, where a significantly higher loss is detected; 3) The scalar value of the loss cannot serve as an quantitative indicator for downstream performance. For example, the variants of *Global RGB Normalization* and *Global Patch Normalization* apply different ways to smooth the inter-patch variances and thus different optimization losses are exhibited. Nevertheless, both of them eventually achieve a similar downstream finetune accuracy; 4) When further appending the intra-patch normalization upon these two variants in 3), their new counterparts are sharing a nearly same loss curve; 5) However, if we directly conduct the intra-patch normalization upon the raw pixels (the green curve in the figure), although it has a similar loss curve with the two variants in 4), the performance of its downstream finetuning is actually much lower.

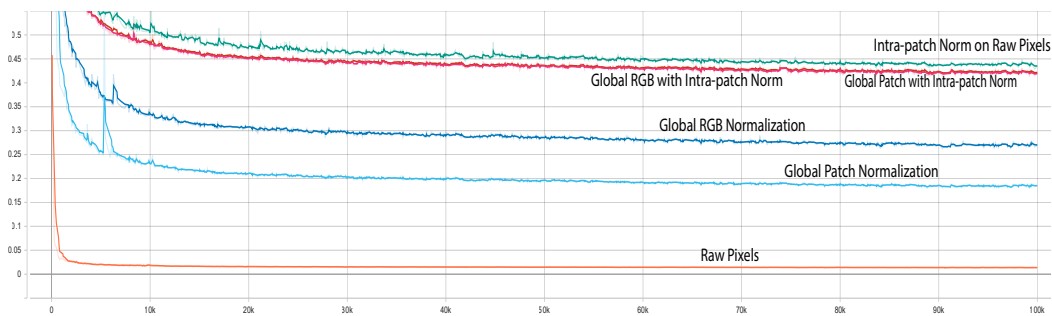

Figure 2: The training loss curves with respect to different reconstructed targets, which are marked with different colors. We conduct this study by pre-training 100 epochs for each variant. The horizontal and vertical axes represent the iteration steps and loss scalars respectively. Refer to the section 3.2 for more detailed observations from this visualization.

Table 3: Performance of transferring the pre-trained models on COCO object detection and segmentation using a ViT Mask R-CNN baseline.

| Method | #PT-Epoch | #FT-Epoch | lr | Box AP | Mask AP |
|---|---|---|---|---|---|
| *Methods using ViT-B/16:* | | | | | |
| MAE (our impl.) | 800 | 25 | 1.0e-4 | 47.7 | 42.6 |
| | 800 | 25 | 1.6e-4 | 48.8 | 43.6 |
| Ours | 800 | 25 | 1.0e-4 | 48.1 | 42.8 |
| | 800 | 25 | 1.6e-4 | 48.9 | 43.7 |
| *Methods using ViT-L/16:* | | | | | |
| MAE (our impl.) | 800 | 25 | 1.5e-4 | 51.9 | 46.0 |
| Ours | 800 | 25 | 1.5e-4 | 52.4 | 46.3 |

Table 4: Performance (mIoU) of transferring the pre-trained models on ADE20K semantic segmentation using UperNet.

| Method | #PT-Epoch | layer_decay | mIoU |
|---|---|---|---|
| *Methods using ViT-B/16:* | | | |
| MAE (our impl.) | 800 | 0.65 | 45.92 |
| Ours | 800 | 0.65 | 46.92 |

## 3.3 DETECTION AND SEGMENTATION

Beyond finetuning on the same benchmark, we also verify the transferring performance of our pre-trained models across different downstream tasks and datasets. First, we conduct object detection and instance segmentation on COCO (Lin et al., 2014) using a ViT Mask R-CNN (He et al., 2017) architecture, which is mainly based on the implementation of Li et al. (2021). Specifically, we finetune the pre-trained model for 25 epcohs using an AdamW (Loshchilov & Hutter, 2017) optimizer with a cosine learning rate scheduler, where the weight decay is set to 0.1. Note that we only tune on the learning rate. If not specified, the other hyperparameters are simply following the recipes of Li et al. (2021). As illustrated in Table 3, our method achieves considerable improvements upon the MAE, especially for the ViT-Large model. It demonstrates that our pre-trained model captures contextual representation across different tokens that universally benefit a set of downstream tasks on varying datasets.

We also apply our pre-trained model into the task of semantic segmentation on ADE20K (Zhou et al., 2017). We conduct the experiments using the UperNet (Xiao et al., 2018) architecture and fine-tune the model for 160K iterations in total. We do not perform the intermediate finetuning like Bao et al. (2021), although it can further improve the mean Intersection over Union (mIoU). The results are shown in Table 4, where our approach obtains superior performance over MAE in terms of the equal pre-training epochs. It demonstrates that our pre-trained models can also benefit those downstream tasks for dense prediction.

## 3.4 FLEXIBLE MASKED TOKEN FOR SMALLER MODEL

Since a 75% masking ratio might be too high for the smaller model (e.g., ViT-Base), in this case study, we design a set of mid-level masked tokens to enhance the token interactions and verify whether it can bring positive gains. Unlike the current practice of MAE to insert back the masked tokens only at the bottleneck between the encoder and decoder, we argue that the incorporation of masked tokens can be more flexible to manipulate the degree of oversmoothing. In particular, we initialize another 25% (i.e., 49) masked tokens and insert them into the encoder at the 8th depth, which encourages more token interactions at the higher layer of the encoder. As illustrated in Table 5, our 800-epoch achieves almost the same performance as the 1600-epoch MAE baseline.

Table 5: Top-1 accuracy (%) on ImageNet by using the mid-level masked token for ViT-B/16.

| Method | #PT-Epoch | Top-1 Accuracy |
|---|---|---|
| MAE (our impl.) | 800 | 83.3 |
| MAE | 1600 | 83.6/83.5 (paper/github) |
| Ours | 800 | 83.5 |

## 4 RELATED WORK

In this part, we first zoom in the specific literatures about masked image modeling and then zoom out to generally introduce the related development of self-supervised learning. After that, we will review several works that inspire this paper, including the approaches to alleviate the posterior collapse in autoencoding and some understandings about the success of recent self-supervised algorithms.

### 4.1 MASKED IMAGE MODELING

To the best of our knowledge, Vincent et al. (2008) is one of the very first works to corrupt the input for autoencoding and then utilize the pre-trained denoising autoencoder to initialize deep neural networks. This philosophy has been widely-adopted in the visual modeling such as the context encoder proposed in Pathak et al. (2016), which models the visual feature learning as to predict a region of the image given its surrounding context. However, for a period of time, this mechanism in the vision community does not stand out since its performance is far behind the supervised counterpart. On the contrary, the similar paradigm has recently reshaped the language modeling with an impressive progress (Kenton & Toutanova, 2019; Liu et al., 2019; Brown et al., 2020), which shifts researchers' attention to exploiting its potential in visual modeling. Thanks largely to the equipment of vision transformer (Dosovitskiy et al., 2020), masked image modeling becomes increasingly promising in the self-supervised visual pre-training. BEiT (Bao et al., 2021) attempts to transfer the practices in BERT correspondingly in the image domain, while the predicted visual tokens are extracted from a pre-trained dVAE (van den Oord et al., 2017; Ramesh et al., 2021). He et al. (2022) removes the randomly masked tokens at the encoding phase and reconstructs their raw pixels using a highly lightweight decoder. Xie et al. (2022b) shares some similar designs with He et al. (2022), such as random masking, lightweight decoder and reconstructing pixels. Wei et al. (2022) demonstrates that the HoG representation can be an alternative option for the predicted target.

### 4.2 SELF-SUPERVISED LEARNING

The regime of supervised learning cannot scale up since human annotations are sometimes very expensive and time-consuming. Self-supervised learning illuminates the dark spaces of artificial intelligence by exploring the limit of unlabeled data. A dominant pipeline of self-supervised learning is to pre-train on large-scale raw data and then adapt the pre-trained model into different downstream tasks. Beyond the masked image modeling, contrastive learning is another dominant paradigm of self-supervised visual pre-training, which pulls closer multiple augmentations of an identical instance while pushes away that of different instances. MoCo (He et al., 2020) creates a dynamic queue to store negative samples and a moving-averaged strategy to maintain consistency between the query and key encoders. SimCLR (Chen et al., 2020) employs an extremely large-batch training to directly provide sufficient negative samples, which also achieves remarkable performance. BYOL (Bootstrap Your Own Latent) (Grill et al., 2020) shows that contrastive learning can still work (and even better) without negative pairs, where the online view is further fed into an extra predictor to predict the projection of the stop-gradient target view. On the other hand, masked signal modeling is not only effective in the image domain, but being a unified pretext task for large-scale pre-training across many domains, including language (Kenton & Toutanova, 2019; Ramesh et al., 2021), video (Tong et al., 2022; Feichtenhofer et al., 2022), audio (Baevski et al., 2020), multi-modal (Bachmann et al., 2022; Aghajanyan et al., 2022), and etc.

### 4.3 Posterior Collapse on Autoencoding

When adapting the pre-trained masked autoencoder into downstream tasks, the decoder will be removed in most of current practices. To this end, similar to all the two-network autoencoding architecture, the masked autoencoder also suffers from the curse of posterior collapse. Informally speaking, the decoder itself might be able to accomplish the reconstruction/translation/generation even though the encoder fails to embed meaningful context from the input. To mitigate this issue, Bao et al. (2021) constructs a discretized token space from a pre-trained codebook. Chen et al. (2022) introduces an alignment constraint to decouple the representation learning and the pretext task completion. Beyond the visual domain, Lu et al. (2021) designs a weak decoder with restricted capacity and attention flexibility to encode better text representations. Zhang et al. (2020) presents a mechanism of token dropping for neural machine translation, expecting the decoding to rely on the source context more heavily.

### 4.4 Understandings of Self-supervised Methods

Some literatures attempting to understand the success of self-supervised algorithms also inspires this paper from various perspectives. Wang et al. (2022) emphasizes the importance of the asymmetric design in self-supervised siamese networks, where the extra predictor can be regarded as an autoencoder, as suggested by Chen & He (2021). In particular, their conclusion (i.e., higher variance in the online branch and lower variance in the target counterpart) is also applicable to the practice of masked image modeling. First, for the online branch, we echo that the whole masked autoencoder is in demand of high-variance interactions across tokens to refrain from the oversmoothing issue. To fulfill this goal, it can randomly mask out a high-portion of tokens at the encoding side, while at the decoding side, it can randomly remove some of the edge connections across different tokens. Second, for the target branch, we demonstrate that the pre-training will benefit from conducting the global normalization on the reconstructed objective to smooth the inter-patch variances, and further followed by an intra-patch normalization. Moreover, the uniformity metric proposed in Wang & Isola (2020) supports us to analyze the token distribution in a spherical space, verifying whether the tokens collapse or not. Xie et al. (2022a) applies some visualization techniques to reveal that the masked image modeling brings strong locality inductive bias to all the layers of the model, when compared with its supervised counterpart.

## 5 Conclusion and Limitation

In this paper, we highlight several ingredients that are of significance for the current success of masked image modeling and propose corresponding techniques to robustly and effectively improve them. First, we argue that the whole autoencoder should circumvent the shortcut interactions across different tokens. To fulfill this goal, the high-portion random masking at the encoding and our proposed edge connection dropout at the decoding turn out be the well-performing practices. Second, we demonstrate that the reconstructed target should be applied with some population statistics to smooth the inter-patch variances. Finally, we show that the introduction of masked tokens can be more flexible, in terms of their inserted position and ratio, to manipulate the token interaction for different models with varying capacities.

However, our current efforts are still driven by the empirical observations. A more rigorous understanding of the success of masked image modeling is expected to be developed. Besides, the proposed techniques are typically applied on a single baseline (i.e., MAE) of masked image modeling for demonstration. In future work, we are looking forward to testifying their efficacy covering more MIM frameworks and downstream tasks.

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
