# OpenReview forum: "A Robustly and Effectively Optimized Pretraining Approach for Masked Autoencoder"
_ICLR.cc/2023/Conference — Submitted to ICLR 2023_

### Official Review · Reviewer_6ZYw · 2022-10-21

**Confidence:** 4
**Clarity, Quality, Novelty And Reproducibility:** The current version of the paper has …
**Correctness:** 2
**Technical Novelty And Significance:** 1
**Empirical Novelty And Significance:** 1
**Recommendation:** 1

**Strength And Weaknesses:**

Weaknesses:
(1) The writing of the paper is a little messy. The authors should highlight their contributions clearly in the Introduction. The proposed method is more like a combination of different components or tricks.
(2) Weak performance improvements. The proposed method only brings slight improvements (about 0.3%) over MAE in downstream classification, detection and segmentation tasks, which is far from sufficient to prove its effectiveness.


**Summary Of The Paper:**

The paper proposes several techniques to benefit the pre-training of MAE. However, it seems like an ensemble of different tricks. The contributions should be further highlighted.

**Summary Of The Review:**

I am inclined to reject this paper since the contribution and experiment problems, please see the weaknesses.

---

### Official Review · Reviewer_GC97 · 2022-10-23

**Confidence:** 4
**Correctness:** 4
**Technical Novelty And Significance:** 1
**Empirical Novelty And Significance:** 1
**Recommendation:** 3

**Clarity, Quality, Novelty And Reproducibility:**

I believe that it would not be hard to reproduce the experiments in this paper. But this paper is lack of novelty, introducing negative impact of the paper quality.

**Strength And Weaknesses:**

Strength: consistent performance gain (over different tasks) achieved by simple modifications (i.e. dropout, position of masked tokens)

Weakness: the major concern is novelty. Although incremental improvement is achieved, adding dropout and moving masked tokens are far from innovation, especially for top conference like ICLR.

**Summary Of The Paper:**

This paper proposes several modifications such as adding dropout, moving the position of masked tokens, which provides incremental performance gain over MAE across vision tasks.

**Summary Of The Review:**

Reject is rated mainly due to the lack of novelty. It is not surprising to achieve incremental improvement by leveraging existing techniques.

---

### Official Review · Reviewer_sWps · 2022-10-24

**Confidence:** 4
**Correctness:** 3
**Technical Novelty And Significance:** 2
**Empirical Novelty And Significance:** 2
**Recommendation:** 3

**Clarity, Quality, Novelty And Reproducibility:**

The definitions of global RGB normalization and global patch normalization in Table 1 are missing, making the contribution difficult to understand at the beginning of this paper.

Section 2.1 is limited to explaining the proposed technique by itself.
The reviewer had to find the ViT paper to understand implementation details.

Figure 1 is hard to understand what the proposed method is.


**Strength And Weaknesses:**

The proposed techniques are reasonable.
However, the reviewer has several concerns.

First, the novelty of the proposed techniques is insignificant.
Dropout and image normalization using per-dataset statistics are well-known approaches to improve accuracy.
A smaller masking ratio for a smaller model is straightforward.

Second, the performance improvement is marginal.
In Table 2, the proposed method improves the Top-1 accuracy by 0.3% from the baseline (MAE (our impl.)).
More importantly, Table 5 presents that the adjusted masking ratio achieves Top-1 accuracy improvement by 0.2%, indicating that the other techniques contribute to only 0.1%.

Third, the clarity of this paper is limited.
The equations for the ablation methods in Figure 2 are not described.
The meaning of MAE (our impl.) with 85.3/.4 (/paper) in Table 2 is unclear.

Lastly, experiments are limited to present the effectiveness of the proposed method.
Ablation studies on pre-training iterations, the proposed techniques, and the masking ratio will be helpful.
Analysis of the relation between image reconstruction performances and downstream task accuracies will be interesting.





**Summary Of The Paper:**

This paper presents three techniques on masked autoencoder (MAE)-based pre-training for downstream tasks.
The first is to apply dropout to the attention modules of ViT.
The second is a modification of the image normalization method from using per-patch statistics to using per-dataset statistics.
The third is to adjust the masking ratio depending on model sizes.
Experimental results present that the proposed method can reduce the pre-training iterations of MAE with comparable accuracies in the downstream tasks.


**Summary Of The Review:**

The proposed three techniques are reasonable to train accurate MAE.
However, the novelty, accuracy improvement on downstream tasks, and experiments are limited.

---

### Official Review · Reviewer_Ukam · 2022-10-31

**Confidence:** 5
**Correctness:** 1
**Technical Novelty And Significance:** 2
**Empirical Novelty And Significance:** 2
**Recommendation:** 3

**Clarity, Quality, Novelty And Reproducibility:**

Clarity: I think I can get the gist of the paper, and I can understand the tables and figures reasonably well. However, I think quite a few more iterations on the writing is definitely helpful (especially with more results), for example to trim the length and the paper more punchy and less verbose.

Quality: It's hard for me to judge the quality in the current format, as the experiments are not yet complete to me (e.g., larger models, fine-tuning results for all the techniques, their compositionally, also showing how it can be called "robust").

Novelty: The main contribution of the paper, in its best format, is to present improved baselines on MAE. Dropout on attention, trying different normalization techniques, revisiting masked tokens in the encoder do not sound too novel to me. However, as the authors also admit upfront, I think it is fine with limited novelty if the goal is to find a better recipe.

Reproducibility: Since the proposed techniques are relatively easy to implement, I would say it is less of a problem to get to the results from another institution, the real problem is to show that the techniques are improving the baseline recipe "significantly" and "robustly".

**Strength And Weaknesses:**

Strengths:
- As an important baseline nowadays for self-supervised learning, improving the recipe of MAE (and potentially can replace the original MAE recipe) is useful and of significance.
- Some explorations here, while minor, are of interest. For example it delves into different ways of normalizing the reconstructing targets and shows their difference. Indeed, it is unclear why normalization can help the down-stream performance.

Weaknesses:
- It is unfortunate, but I have to say that after reading the paper, I believe a reasonable user of MAE will still use the original MAE recipe. So the primary goal of the paper is not achieved. There are many reasons for this, for example: 1) the final performance is still not beating MAE significantly. There could be some reproducibility issues, but to claim "robust and effective", just with improvements that close the "reproducibility" gap is definitely not enough. 2) Right now the experimental section is not complete, e.g., I could not find the influence of final ImageNet accuracy with different ways or normalization, so this means it is not conclusive. 3) In order to show the recipe is indeed robust, I would very much like to see its effect on even larger models (ViT-H). 4) How robust is the technique in terms of compositionally?
- I can tell that the paper is done in a rush. The experiments are not yet complete, there are quite a few broken (and lengthy) sentences, and lots of space is devoted to less important things (I believe for a report like this, an introduction should just be half a page long -- 1.5 pages are too long and way more space can be devoted to experiments if they are done.

**Summary Of The Paper:**

The submission manifests several tricks in order to potentially improve the original MAE's recipe. The tricks include: 1) addition dropout in the attention layers; 2) a study on the normalization targets; and 3) intermediate mask tokens. The results are slightly improved with the authors' own implementation.

**Summary Of The Review:**

Overall, this is not yet a complete work to me. After reading the paper, I don't think as a reasonable reader, one would be excited enough to switch the recipe from the original MAE one to the current one. I think it would be great for the authors to think from this perspective, and show more "exciting" results, findings etc. that can actually achieve the goal of replacing the baseline MAE. Therefore, I am firmly on the rejection side and would recommend a "robustly" and "effectively" improved version could be submitted to future venues.

---

### Decision · Program_Chairs · 2023-01-20

**Decision:**

Reject

**Justification For Why Not Higher Score:**

- Modest empirical improvements using existing techniques, limited empirical study
- Lack of novelty, lack of clarity in writing

**Justification For Why Not Lower Score:**

N/A

**Metareview: Summary, Strengths And Weaknesses:**

The authors present and empirically analyze several techniques for improving masked-autoencoding based pretraining. Reviewers pointed out several critical issues with this work, including novelty, clarity, and significance. Given the modest empirical improvements the reviewers unanimously agree that virtually all practitioners will still apply the original MAE training recipe and consider this work not ready for publication at ICLR.